# Release of High-Mobility Group Box-1 after a Raynaud’s Attack Leads to Fibroblast Activation and Interferon-γ Induced Protein-10 Production: Role in Systemic Sclerosis Pathogenesis

**DOI:** 10.3390/antiox12040794

**Published:** 2023-03-24

**Authors:** Yehya Al-Adwi, Isabella M. Atzeni, Berber Doornbos-van der Meer, Amaal Eman Abdulle, Anniek M. van Roon, Alja Stel, Harry van Goor, Andries J. Smit, Johanna Westra, Douwe J. Mulder

**Affiliations:** 1Department of Internal Medicine, Division of Vascular Medicine, University Medical Centre Groningen, University of Groningen, 9700 RB Groningen, The Netherlands; y.a.s.mohammed@umcg.nl (Y.A.-A.);; 2Department of Rheumatology and Clinical Immunology, University Medical Centre Groningen, University of Groningen, 9700 RB Groningen, The Netherlands; 3Department of Pathology and Medical Biology, Section Pathology, University Medical Centre Groningen, University of Groningen, 9700 RB Groningen, The Netherlands

**Keywords:** injury, ischemia and reperfusion, Raynaud’s phenomenon, SSc, IP-10, HMGB1, RAGE

## Abstract

Raynaud’s Phenomenon (RP) leading to repetitive ischemia and reperfusion (IR) stress, is the first recognizable sign of systemic sclerosis (SSc) leading to increased oxidative stress. High-mobility group box-1 (HMGB1) is a nuclear factor released by apoptotic and necrotic cells after oxidative stress. Since HMGB1 can signal through the receptor for advanced glycation end products (RAGE), we investigated whether an RP attack promotes the release of HMGB1, leading to fibroblast activation and the upregulation of interferon (IFN)-inducible genes. A cold challenge was performed to simulate an RP attack in patients with SSc, primary RP (PRP), and healthy controls. We measured levels of HMGB1 and IFN gamma-induced Protein 10 (IP-10) at different time points in the serum. Digital perfusion was assessed by photoplethysmography. In vitro, HMGB1 or transforming growth factor (TGF-β1) (as control) was used to stimulate healthy human dermal fibroblasts. Inflammatory, profibrotic, and IFN-inducible genes, were measured by RT-qPCR. In an independent cohort, sera were obtained from 20 patients with SSc and 20 age- and sex-matched healthy controls to determine HMGB1 and IP-10 levels. We found that HMGB1 levels increased significantly 30 min after the cold challenge in SSc compared to healthy controls. In vitro stimulation with HMGB1 resulted in increased mRNA expression of IP-10, and interleukin-6 (IL-6) while TGF-β1 stimulation promoted IL-6 and Connective Tissue Growth Factor (CTGF). In serum, both HMGB1 and IP-10 levels were significantly higher in patients with SSc compared to healthy controls. We show that cold challenge leads to the release of HMGB1 in SSc patients. HMGB1 induces IP-10 expression in dermal fibroblasts partly through the soluble RAGE (sRAGE) axis suggesting a link between RP attacks, the release of HMGB1 and IFN-induced proteins as a putative early pathogenetic mechanism in SSc.

## 1. Introduction

Systemic sclerosis (SSc) is a systemic disease that involves a triad of chronic complications including autoimmunity, vasculopathy, and inflammatory fibrosis of the skin and internal organs [1]. The pathogenesis of SSc is complicated and little is known about the initiating factors, especially in early disease. Therefore, there is an unmet need to study early (molecular) mechanisms to identify potential treatment targets. Microangiopathy is one of the earliest complications of SSc, mostly represented by Raynaud’s phenomenon (RP). Around 95% of diagnosed SSc patients suffer from RP which is also considered a hallmark of early SSc disease. RP is an exaggerated vasospastic disorder of the extremities, leading to repetitive ischemia and reperfusion stress [2]. The latter is considered a contributor to organ damage through oxidative stress mechanisms leading to the release of danger-associated molecular patterns (DAMPs) including, among others, advanced glycation end products (AGEs) and reactive dicarbonyl compounds, calgranulins (S100A8/9/12) and high mobility group box 1 (HMGB1) [3,4,5,6].

Several DAMPs signal through the receptor for advanced glycation end products (RAGE) and toll-like receptor 4 (TLR4), and HMGB1 is no exception. HMGB1, a proinflammatory DNA-binding nuclear cytokine, is released due to cell injury by activated, necrotic and apoptotic cells [7]. A recent study has shown that both the soluble form of RAGE (sRAGE) and its ligand, HMGB1 are elevated in the serum and skin of patients with SSc [8]. The downstream processes of the HMGB1-RAGE/TLR4 axis are proinflammatory and mostly signal through the type I interferon (IFN-I) axis [9,10,11].

Inflammatory mechanisms are suspected to initiate SSc complications in early disease [12]. Part of these mechanisms are governed by type-I T helper cell (Th1) cytokines such as interleukin-2 (IL-2), IL-12, IL-23 and others which lead to the activation of reaction cascades that are essential for the trafficking and maturation of effector immune cells [13,14]. Recruited and matured cells will release IFN-I proteins and chemokines which, in turn, recruit inflammatory cells such as monocytes, CD8+ T, natural killer, and dendritic cells [15,16]. One important chemokine that is involved in the previous process is Interferon-γ-induced protein 10 (IP-10, also known as CXCL10). IP-10 is an IFN-induced chemokine that is also a major pro-inflammatory chemokine that functions through binding to C-X-C chemokine receptor 3 (CXCR3)-expressing cells [17]. Serum IP-10 is known to be elevated in SSc patients compared to healthy controls and has been reported to be at its peak levels in early disease [18,19].

Previously, we have shown that an RP attack led to an increased release of free thiols in SSc patients compared to healthy controls and primary Raynaud patients [20]. Not only does this reflect a balanced redox system but also could mean increased oxidative stress in this particular group which requires a higher release of antioxidants, such as free thiols. In this work, we investigated whether an RP attack, as an early manifestation of SSc disease and a clinical sign of high oxidative stress, would promote the release of HMGB1. Then, we performed in vitro experiments to unravel whether HMGB1, in turn, would lead to the production of IP-10 through the sRAGE axis in dermal fibroblasts. Finally, we measured serum HMGB1 and IP-10 in an independent cohort to check whether levels of HMGB1 and IP-10 are also increased chronically in SSc patients compared to healthy populations.

## 2. Materials and Methods

The design of this study is based on (induced) clinical observations and then aims to replicate (some of) these observations in vitro to understand the outcomes. We first investigated whether inducing a Raynaud attack would differ in SSc patients compared to healthy volunteers by measuring reperfusion and release of HMGB1 before and after the induced attack. Then, we used HMGB1 to induce skin fibroblasts where we measured the gene expression of relevant proinflammatory and fibroproliferative genes. Lastly, in an independent cohort, we measured levels of HMGB1 and IFN-induced protein in sera of SSc patients and healthy controls. 

Patient involvement
Three groups of study subjects have been involved in Raynaud’s attack induction study. The first group comprised 10 healthy controls with no history of RP. The second group comprised 10 PRP patients (primary Raynaud’s phenomenon with the absence of underlying SSc disease) while the third group was composed of 10 SSc patients with RP. All SSc patients met the American College of Rheumatology/European League Against Rheumatism 2013 (ACR/EULAR 2013) criteria [21]. The diagnosis of primary RP was confirmed by the treating physician and patients had no SSc-related autoantibodies and absence of aberrant nail-fold capillaries and digital ulcers. The independent cohort comprised two groups of sex- and age-matched study subjects: SSc patients (*n* = 20) and healthy controls (*n* = 20). This cohort (Cohort B) is completely independent of the induced Raynaud’s attack cohort (Cohort A).

All study subjects refrained from smoking on the day blood was drawn and vascular measurements were performed.

Blood samples were obtained, and serum was stored at −20 °C. Written informed consent was obtained from all study subjects prior to participation and the study was approved by the Local Medical Ethics Committee of the University Medical Center Groningen (A: METc 2015/219, B: 2014/337).

Photoelectric plethysmography (PPG) of the fingers (Raynaud’s induction)

The cooling and recovery experiment was performed on each subject according to a standardized protocol as described previously [20]. Subjects were asked not to drink or smoke for 8 h prior to the experiment. Before, during and after cooling, perfusion of the five digits of the left hand was measured and recorded by PPG as described previously [22,23]. Briefly, PPG was recorded, and blood was drawn after the subject has been acclimatized in the room (23–24 °C) for 30 min (T0). Thereafter, the subject’s left hand was submerged in preheated water (33 °C) for a duration of 36 min where water was cooled down until 6 °C with PPG measurements every 4 min. After 36 min of cooling or until unendurable pain, blood was drawn (T1). The hand of the subject was left to rewarm at room temperature for 30 min with two-time points for blood drawing: 10 and 30 min after the start of rewarming (T2 and T3, respectively). Mean ischemic time (min) was calculated for all fingers and was defined as the mean time of perfusion loss during cooling and recovery. PPG pulses were calculated from R-peak. The RP was deemed “induced” when ≥2 fingers lost perfusion and stayed abnormal during ≥2 consecutive steps. Data acquisition was performed using a Biopac MP-100 system with five PPG100C amplifiers and PPG200C sensors (infrared light, 860 nm), an ECG100C amplifier with ECG-cables, an SKT100C amplifier and TSD202C temperature sensor, and AcqKnowledge 3.8.2 software (Biopac Systems Inc., Goleta, CA, USA).

Enzyme-linked immunosorbent assay (ELISA)

After blood was drawn from the research subjects into SST Vacutainer tubes (BD), tubes were centrifuged at 1300× *g* for 10 min to separate serum and stored at −20 °C. After thawing, levels of HMGB1 were measured in sera of subjects who underwent Raynaud’s induction experiment and in the independent cohort using a commercially available ELISA kit (TECAN, Hamburg, Germany) according to manufacturer instructions. IP-10 was measured in sera of the independent cohort using Duoset ELISA kit (R&D systems, Minneapolis, MN, USA). High-performance ELISA buffer (Sanquin, Amsterdam, The Netherlands) was used during serum incubation to prevent non-specific reactions. IL-6, CTGF and IP-10 protein levels were measured using Duoset ELISA kits (R&D systems, Minneapolis, MN, USA) in the supernatant of HMGB-1 and TGF-β-stimulated dermal fibroblasts (72 h stimulation) and similarly with fibroblasts that were treated with the RAGE inhibitor before stimulation.

Dermal fibroblasts culture and treatment

Primary human dermal fibroblasts obtained from mammal resection were cultured in Dulbecco**’**s modified Eagle**’**s medium ([DMEM] high glucose, Biowest, MO, USA), and supplemented with 10% Fetal Calf Serum (FCS), 1% penicillin/streptomycin. Cells were incubated at 37 °C (5% CO_2_, 95% oxygen) and passaged when 90% confluent. Cells were used at passage numbers 4–9. They were seeded in 6-well plates at 0.5 × 10^6^ cells per well. Before treatment, cells were quiesced for 24 h. Then, the medium was discarded, and cells were stimulated with 2 µg/mL HMGB1 (Bovine HMGB1, Chondrex, WA, USA) or 10 ng/mL recombinant human tissue growth factor-β1 (TGF-β1, Peprotech, NJ, USA) for 6 h or 72 h (for protein studies). Furthermore, to investigate the role of RAGE, a specific RAGE inhibitor FPS-ZM1 (EMD Millipore, Molsheim, France) was added at a concentration of 10 µM 30 min before stimulation with TGF-β1 and HMGB1. Negative control wells were treated with culture media only. Fibroblasts from each well were harvested into a 1.5 mL Eppendorf cup using 1 mL of TRIzol (Invitrogen, CA, USA). Finally, the samples were stored at −80 °C until further analysis.

Reverse transcriptase-quantitative polymerase chain reaction (RT-qPCR) analysis

Total RNA was extracted according to the manufacturer’s instructions. After extraction, DNA was digested using DNAse enzyme. Thereafter, RNA quantity and quality were assessed using NanoDrop 1000 spectrophotometer (Thermo Scientific, Wilmington, DE, USA). RNA sample quality (260/280) >1.8 was sufficient to proceed with the analysis. Thereafter, 25 μL of RNA samples of max 100 ng RNA were utilized to synthesize cDNA according to manufacturer instructions. After cDNA synthesis, 1 uL of each sample was pipetted (in duplicate) into a 384-well plate. Thereafter, mRNA levels of interleukin-6 (IL-6), collagen-1α (Col-1α), connective tissue growth factor (CTGF), α-smooth muscle actin (α-SMA), interferon genes (IFN α-induced 44L [IFI44L], Myxovirus resistance protein 1 [Mx1], Lymphocyte antigen 6 complex, locus E [LY6E], and interferon-γ-inducible protein-10 [IP-10]), and glyceraldehyde-3-phosphate dehydrogenase (GAPDH) were measured by an Applied Biosystems™ QuantStudio™ 6 Flex Real-Time PCR System (Singapore) with specific TaqMan assays (IL-6 [Hs00174131_m1], α-SMA [ACTA2, Hs00909449_m1], Col-1α [Hs00164004_m1], CTGF [Hs00170014_m1], IFI44L [Hs00915292_m1], and IP-10 [Hs00171042_m1], Applied Biosystems, Warrington, UK). The amount of target was normalized to an endogenous reference (GAPDH [Hs99999905_m1]), and expressed as relative expression (2–ΔCT) or as fold induction compared to an unstimulated sample (2-ΔΔCT). Data were analyzed using QuantStudio Real-Time PCR software v1.3 (Applied Biosystems, Singapore).

Statistical analysis

SPSS version 22 (IBM, Chicago, IL, USA) was used to perform statistical analysis. PPG data were presented as mean ± standard deviation. Other non-normally distributed data were depicted as median (interquartile range). Differences between groups were tested using Kruskal–Wallis and when significant, Mann–Whitney U is used). Logistic regression models were built to correct for possible confounders. A *p*-value < 0.05 was considered significant.

## 3. Results

### 3.1. Patient Characteristics

Original inclusion found 32 eligible subjects; however, we had to exclude two subjects due to the unavailability of PPG measurements. The clinical characteristics of the 30 study subjects can be found in Table 1. The median age was 43 years (18–73) and two-thirds of the subjects were females (HC = 7, PRP = 8, SSc = 5). Importantly, we were able to induce Raynaud attack in all PRP and SSc subjects but not in healthy controls. The premature ending of the cooling experiment due to unendurable pain was carried out in three PRP and seven SSc subjects. The independent cohort consisted of 40 subjects; SSc patients and age- and sex-matched healthy controls (50:50). The clinical characteristics of the study subjects can be found in Table 2. Logistic regression analysis showed that no confounders could have influenced the HMGB-1 or IP-10 measurements except for every smoker status (*p* = 0.057 and *p* = 0.43, respectively). To be more specific, the analysis showed that every smoker status could have significant independent effects on HMGB-1 measurement (*p* = 0.039). However, when correcting using the number of packyears, the analysis showed that smoking had no independent effects (*p* = 0.339) on HMGB-1 (*p* = 0.032) or IP-10 (*p* = 0.029) levels between groups.

**Table 1 antioxidants-12-00794-t001:** Clinical characteristics of subjects who underwent the cooling experiment.

	Healthy Controls (HC) *n* = 10	Primary Raynaud Phenomenon (PRP) *n* = 10	Systemic Sclerosis (SSc) *n* = 10
Age (years), mean ±SD	29.9 ± 2.7	47.8 ± 16.2	56.2 ± 11.5
Female gender, *n*	7	8	5
Disease duration (years), median (IQR)	N/A	8.0 (4.0–17.0)	4.5 (3.0–6.3)
Abnormal NCM pattern, *n*	0	0	10
Positive autoantibodies, *n*	0	0	8
Organ involvement, *n*			
Pulmonary	0	0	0
Oesophageal	0	0	3
Comorbidities			
Diabetes, *n*	0	1	0
Hypertension, *n*	0	1	2
Hypercholesteremia, *n*	0	0	1
Creatinine, median (IQR)	72.1 [66.5–77.6]	72.0 [63.5–72.0]	76.2 [63.5–88.6]
eGFR, median (IQR)	94.5 [90.6–114.8]	79.3 [68.3–97.4]	86.6 [74.1–98.5]
History of Digital ulcers, *n*	0	0	6
Relevant medication, *n*			
Immunosuppressants			
Plaquenil	0	0	1
Mycophenolate mofetil	0	0	1
Calcium channel blockers			
Nifedipine	0	1	2
Vasodilators			
Iloprost	0	0	2
Bosentan	0	0	3
Other antihypertensives	0	1	3
Lipid-lowering drugs	0	0	1
Glucose-lowering drugs	0	1	0
Anticoagulants	0	1	2

N/A: Not Available; NCM: Nailfold capillary microscopy; eGFR: estimated glomerular filtration rate.

**Table 2 antioxidants-12-00794-t002:** Clinical characteristics of the independent cohort.

	Healthy Controls (*n* = 20)	SSc Patients (*n* = 20)
Female, *n* (%)	14 (70.0)	13 (65.0)
Age in years, median (IQR)	52 (45–62)	51 (44–58)
Caucasian ethnicity, *n* (%)	19 (95.0)	18 (90.0)
Smoker (ever), *n* (%)	7 (35.0)	15 (75.0)
RP duration in years, median (IQR)		8.5 (4.3–13.0)
Disease duration since first non-RP symptom in years, median (IQR)		2.0 (1.0–7.8)
Organ involvement, *n* (%)		
Lung		10 (50.0)
PAH		1 (5.0)
ILD		9 (45.0)
Gastrointestinal		8 (40.0)

RP: Raynaud’s Phenomenon; PAH: Pulmonary Arterial Hypertension; ILD: Interstitial Lung Disease.

### 3.2. Digital Perfusion Recovery Is Slower in SSc Patients after Raynaud’s Attack

We induced a Raynaud’s attack in the three groups and measured the normal perfusion percentage of all digits during cooling and recovery using PPG (Figure 1A). We noticed a longer mean ischemic time in SSc patients (30 min, range: 24–37 min) compared to PRP and healthy controls (12 min, range: 9–14 min, *p* = 0.01, and 1.1 min, range = 0–3.7 min, *p* < 0.001, respectively). Additionally, SSc patients had significantly longer recovery time compared to the PRP group (8 min, range = 4–10 min, and 1.1 min, range = 1.0–2.1 min, *p* < 0.001, respectively) (Figure 1B).

### 3.3. Cold Stress in SSc Patients Leads to the Release of HMGB1 

We took blood samples at three different time points before, 10 and 30 min after the cold challenge and measured the levels of HMGB1 and IP-10 in the sera of the subjects of the three groups. Results indicated that HMGB1 levels in ssc patients were significantly increased after 30 min of the cold challenge compared to healthy controls (*p* < 0.01), but not in PRP subjects (Figure 2A). Levels of IP-10 did not change throughout the experiment (Figure 2B). 

### 3.4. HMGB1 Stimulation of Dermal Fibroblasts Induce Inflammatory Gene and Protein Expression

Since HMGB1 levels were elevated in SSc patients after a Raynaud’s attack, we sought its effects on normal primary human dermal fibroblasts. We investigated proinflammatory (IL-6), profibrotic (Col-1, CTGF and a-SMA) and IFN (IFI44L, Mx1, IP-10 and Ly6e) gene expressions after stimulating the fibroblasts and found that IL-6 and IP-10 gene expression was induced 3- and 20- folds compared to controls, respectively. Expression of the other genes that were included in the panel was changed less than two-fold (Figure 3A). We also found that inhibiting RAGE signaling using FPS-ZM1 led to a decrease in the induced gene expression of IL-6 and IP-10 by 50% (Figure 3B). In order to better understand whether HMGB1 stimulates skin fibroblast in an inflammatory fashion that is partly governed by sRAGE signaling, we performed the same experiments while using TGF-β, a potent fibrotic cytokine, instead of HMGB1 and analyzed the same gene panel. Skin fibroblasts overexpressed IL-6, Col-1, CTGF and IFI44L (13-, 4-, 33- and 10-fold, respectively) compared to controls (Figure 3C). Using RAGE inhibitors did not affect the overexpression of the mentioned genes (Figure 3D) as when we used it with HMGB1 stimulation. Furthermore, we investigated the protein levels of the most relevant and differentially overexpressed genes (IP-10, CTGF, and IL-6). We found that both IL-6 and CTGF protein levels in the supernatant of the stimulated cells are in concordance with gene levels while we were not able to detect any IP-10 levels.

### 3.5. HMGB1 and IP-10 Levels in SSc Patients (Independent Cohort)

Since HMGB1 was elevated in SSc patients after induced Raynaud’s attack and IP-10 was overexpressed in dermal fibroblasts treated with HMGB1, we sought whether HMGB1 (Figure 4A) and IP-10 (Figure 4B) levels are consistently elevated in SSc patients. In an independent cohort, we measured HMGB1 and IP-10 serum levels and found that both proteins are significantly elevated in SSc patients compared to control (*p* = 0.028 and *p* = 0.009, respectively). 

## 4. Discussion

Systemic sclerosis is a complex heterogeneous disease with several proposed pathogenesis pathways including a clear role for oxidative stress. Unraveling the key factors in the initiation of the disease would potentially allow future interventions. Inflammatory events are suspected to govern the early phase of SSc. In this translational study, we have shown that only definitive SSc patients significantly increased HMGB1 release (or secretion) after a Raynaud attack compared to PRP and HCs. Then, we opted for an in vitro model to understand the inflammatory/fibrotic effects of HMGB1 on skin fibroblasts and found that inflammatory genes are induced upon HMGB1 stimulation with a strikingly increased expression of IP-10 which was suppressed when we blocked the RAGE-axis. In an independent cohort, we confirmed that serum levels of HMGB1 and IP-10 in SSc patients were significantly elevated compared to healthy controls.

Although HMGB1 has important physiological functions, it has been attributed to several pathological conditions mainly because it could function as a DAMP molecule when released extracellularly [24]. There are several possible mechanisms for the release of HMGB1 in SSc patients, one is oxidative stress [25]. We have shown in a previous study that SSc patients after a Raynaud attack produce far more antioxidants (free thiols) than PRP and HCs [20]. This suggests that there are initiated damage processes in SSc patients which, in turn, can lead to the release of DAMPs including HMGB1. Additionally, studies have shown that nuclear factor-erythroid 2-related factor2 (NRF2), a potent antioxidant transcription factor, inhibits the release of HMGB1 through the induction of Heme oxygenase-1 (HO-1) [25,26]. In SSc patients, skin fibroblasts expressed NRF2 and HO-1 at a significantly lower level than in fibroblasts from healthy controls [27]. This can be interpreted as that patients with SSc have disrupted antioxidative abilities which may lead to higher damage events and release of HMGB1 as a result. 

As in many autoimmune diseases, SSc patients are often characterized by disease-specific autoantibodies including anti-scl70, anti-centromere, anti-RNA polymerase III autoantibodies and others [28]. Although these autoantibodies have been associated with specific clinical manifestations and used as prognostic biomarkers, little is known about their pathogenetic mechanisms [29]. Recently, Raschi et al. [30] have studied the pathogenetic effects of SSc-specific autoantibodies on endothelial cells. They have demonstrated that these autoantibodies, especially anti-scl70 and anti-centromere, embedded in immune complexes (Abs-ICs) damage the endothelium. Specifically, Abs-ICs induced proinflammatory and profibrotic signatures in healthy cultured endothelial cells. In the same context, Servettaz et al. demonstrated a direct role of reactive oxygen species (ROS) in SSc development in an SSc mouse model. Importantly, they revealed that the type of ROS, which consequently dictates the oxidation pattern of the autoantibody, influences the type of clinical manifestations SSc patients can develop [31]. The relevance of these data to what we reported here is that the reported endothelial damage is one of the earliest events in SSc represented by RP and while this damage has clinical presentation, it also may be connected to mechanistic pathogenesis including HMGB1 release as seen in our study, and as has been reported in lupus vasculitis [32]. Additionally, our in vitro model showed that the fibroblasts (fibrosis-effector cells) can play an inflammatory role in early SSc disease. Frequently, fibroblasts are thought to be only involved in fibrotic, late-stage SSc. Here, we shed a light on a parallel possible mechanism, to endothelium activation, that could lead to disease progression in early SSc events. We demonstrated that skin fibroblasts are responsive to inflammatory stimuli and capable of producing key inflammatory genes and proteins.

Our mechanistic in vitro studies show that HMGB1 induces skin fibroblasts to become more inflammatory by inducing type-1 IFN and IL-6 expression partly through a RAGE-dependent pathway. These observations are in line with early disease inflammatory manifestations. TGF-β is known to be one of the most potent profibrotic cytokines and several studies in different diseases have shown its fibrotic effects in skin fibroblasts [33]. It also mediates tissue remodeling and fibrosis in SSc when the disease progresses to a more fibrotic phenotype [34]. Here, we stimulated the skin fibroblasts with TGF-β as a positive control to show the differences in inflammatory, fibrotic and IFN-induced gene expressions compared to HMGB1 stimulation. Overexpression of profibrotic genes was more predominant in the case of TGF-β compared to HMGB1 stimulation. Additionally, the inhibitory studies using FPS-ZM1 implicate that HMGB1 signals through the RAGE-axis to a great extent. IP-10 and IL-6 expression was decreased after RAGE-axis inhibition by more than 50%. This culminates the importance of the HMGB1-RAGE axis in early inflammatory processes in SSc and could be a potential pathway to therapeutically target to suppress early inflammation.

Interestingly, IP-10 showed a striking increase (20-fold) after HMGB1 stimulation. As mentioned, IP-10 is a major pro-inflammatory chemokine that is involved in the pathophysiology of multiple autoimmune diseases. When the IP-10/CXCR3 axis is activated, it initiates a positive feedback loop of Th1 cell recruitment perpetuating the inflammatory reaction cemented by more IFN-γ release. Taken together, our cooling experiment showed that SSc patients had prolonged ischemia after RP attack compared to PRP and healthy controls. Additionally, only SSc patients showed an increase in HMGB1 but not IP-10 levels after 30 min of recovery (indirect/room air warming). When we treated skin fibroblasts with HMGB1, IP-10 expression strikingly increased while it decreased upon using a RAGE-axis inhibitor. This possibly means that HMGB1 released during RP episodes binds to RAGE which leads to overexpression of IP-10 which, in turn, chemoattracts inflammatory cells sustaining low-grade chronic inflammation which is the hallmark of early SSc disease. Moreover, the interpretations of our data are in line with results from a longitudinal study where newly diagnosed SSc patients had higher levels of IP-10 which declined on follow-up [18].

By default, patients with SSc have higher IP-10 compared to the healthy population [19]. In our independent cohort not only did we show higher IP-10 levels but also demonstrated that HMGB1 levels are higher in patients with SSc compared to healthy controls. Such observations along with the rest of the study support the notion that the HMGB1-RAGE-IP-10 axis is one main pillar in sustaining the early inflammatory phase in patients with SSc. Although speculative, since our data from the cooling study showed higher HMGB1 release in SSc patients and the in vitro study showed higher IP-10 release after HMGB1 stimulation, HMGB1 and/or IP-10 serum concentrations might be beneficial as an early biomarker for elevated oxidative stress/tissue damage in SSc patients which warrant intervention. The HMGB1-RAGE axis seems to be an attractive therapeutic target for early SSc that need to be explored in future studies [35].

Despite the potential of this exploratory study, it has several limitations. The small sample size hindered this study from giving definitive conclusions. Additionally, not all SSc patients were at the early inflammatory phase, yet the study helped in understanding potential pathogenesis pathways using a unique validated experiment modeling a very early event of SSc. It is important to understand that we have not studied all the DAMPs (AGEs, calgranulins, etc.) and receptors (TLR4, TLR9, etc.) that can be involved in I/R stress which could also be involved in the damaging process. Our in vitro model included skin fibroblasts only; we have not investigated other potential cells such as endothelial cells, dendritic cells and macrophages. However, fibroblasts are the fibrosis-effector cells that when stimulated hyperproliferate and produce ECM cues leading to fibrosis. Additionally, other studies have shown the potential of the mentioned cells which have been partly discussed here. We could not detect IP-10 protein levels in the supernatant of stimulated fibroblasts. This might be due to the limit of the detection of the IP-10 ELISA or the fact that IP-10 production is, by default, low. The mRNA relative expression of IP-10 was indeed relatively low compared to the other genes which, in turn, could explain too low protein levels to be detected by ELISA. Finally, while we have focused on some IFN-induced proteins, especially IP-10, it is worth mentioning that there are many other IFNs and IFN-induced proteins that were not discussed here. We focused on IP-10 due to its relevance in early SSc disease, and its potential to be used as a biomarker and as an attractive therapeutic target. 

## 5. Conclusions

This translational study demonstrates that induced IR/oxidative stress represented by RP attack in patients with SSc leads to the release of HMGB1. In vitro, HMGB1 induced IP-10, partially, in a RAGE-dependent manner. This links HMGB1 release following an RP attack to interferon-inducible proteins as putative sequential steps leading to disease progression in SSc. It also paves the way for therapeutic entities to target the HMGB1-RAGE-IP-10 axis. Future research should seek other (inflammatory) pathways, possibly TLRs pathways, which could complement the mentioned pathway and find possible interventions.

## Figures and Tables

**Figure 1 antioxidants-12-00794-f001:**
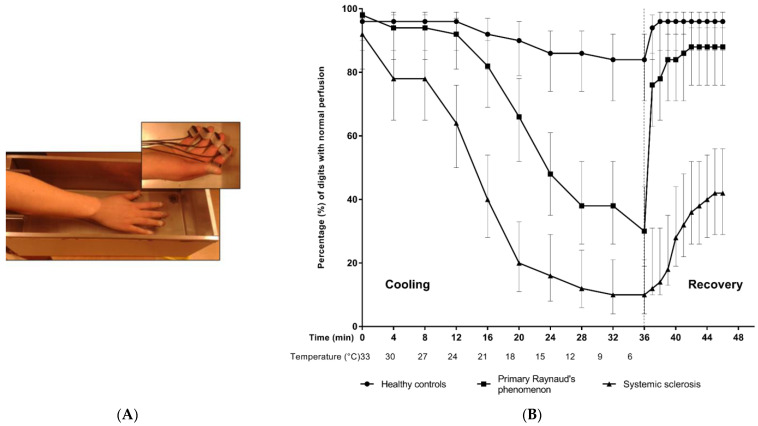
Digital perfusion during cooling and recovery experiment. (**A**) Shows a snapshot of the cooling experiment while the hand is submerged in water to perform the cold challenge and digitals are wired to the photoelectric plethysmographic sensors. (**B**) Percentage of digits with normal perfusion (95% CI) assessed with photoelectric plethysmography during stepwise cooling and recovery. Lines represent healthy controls (*n* = 10), patients with primary Raynaud (*n* = 10) and SSc patients (*n* = 10). The vertical dashed line indicates the beginning of the recovery phase. All healthy controls completed the entire cooling duration of 36 minutes. Seven primary Raynaud patients completed the cooling experiment, two patients discontinued after 32 minutes (9 °C), and one patient discontinued after 12 minutes (24 °C). In addition, three systemic sclerosis patients completed the cooling experiment, one patient discontinued after 20 minutes (18 °C), one patient discontinued after 24 minutes (15 °C), 2 patients discontinued after 28 minutes (12 °C), and three patients discontinued after 32 minutes (9 °C). In addition to the cooling phase, the perfusion of all participants was measured during the recovery phase. HC: healthy controls; PRP: primary Raynaud’s phenomenon; SSc: systemic sclerosis. The figure is adapted from [20].

**Figure 2 antioxidants-12-00794-f002:**
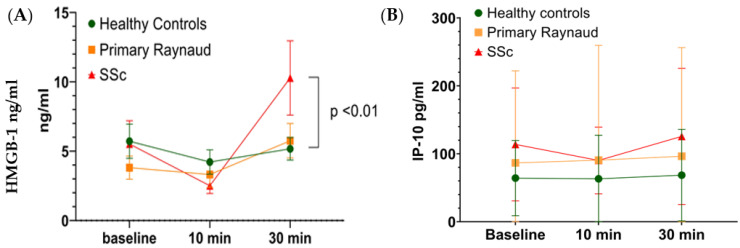
HMGB1 and IP-10 levels before and after the cold challenge. (**A**) HMGB1 levels in healthy controls, primary Raynaud’s subjects and SSc patients at baseline, 10 and 30 mins after cold challenge. Each point represents the mean concentration in all subjects of each group. HMGB1 levels are depicted in ng/mL. (**B**) IP-10 levels in healthy controls, primary Raynaud’s subjects and SSc patients at baseline, 10 and 30 mins after cold challenge. Each point represents the mean concentration in all subjects of each group. IP-10 levels are depicted in pg/mL. SSc: Systemic sclerosis. High mobility group box 1 (HMGB1), interferon-γ-inducible protein-10 (IP-10).

**Figure 3 antioxidants-12-00794-f003:**
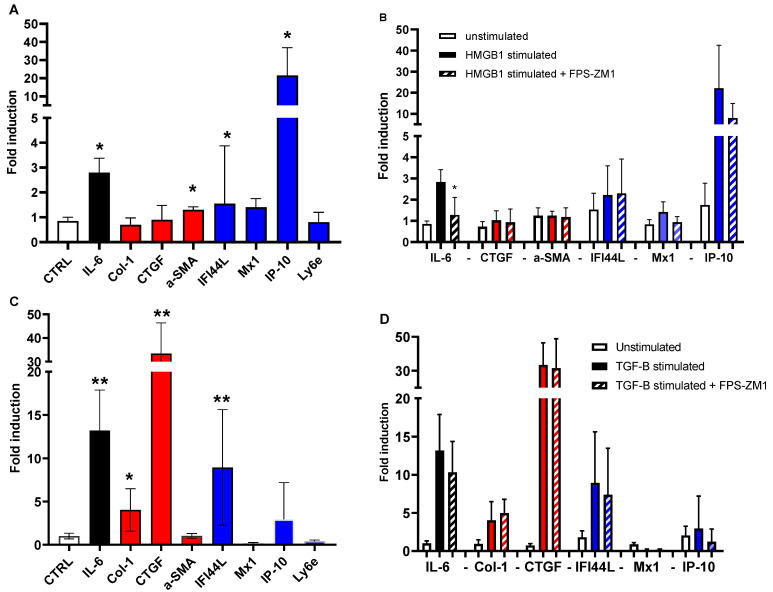
Gene expression of proinflammatory, profibrotic, and interferon genes in HMGB1 or TGF-β stimulated dermal fibroblasts. (**A**) Dermal fibroblasts were stimulated with HMGB-1 and gene expression of IL-6, Col-1, CTGF, A-SMA, IFI44L, Mx1, IP-10 and Ly6e were investigated. (**B**) a RAGE inhibitor (FPS-ZM1) was used before HMGB-1 stimulation and IL-6, IFI44L, Mx1 and IP-10 gene expression was analyzed. (**C**) Dermal fibroblasts were stimulated with TGF-β and gene expression of IL-6, Col-1, CTGF, A-SMA, IFI44L, Mx1, IP-10 and Ly6e were investigated. (**D**) a RAGE inhibitor (FPS-ZM1) was used before TGF-β stimulation and IL-6, IFI44L, Mx1 and IP-10 gene expression was analyzed. Statistical significance: * *p* < 0.05, ** *p* < 0.01. High mobility group box 1 (HMGB1), transforming growth factor β (TGF-β), Interleukin-6 (IL-6), collagen-1α (Col-1α), connective tissue growth factor (CTGF), α-smooth muscle actin (α-SMA), IFN α-induced 44L (IFI44L), Myxovirus resistance protein 1 (Mx1), Lymphocyte antigen 6 complex, locus E (LY6E), and interferon-γ-inducible protein-10 (IP-10).

**Figure 4 antioxidants-12-00794-f004:**
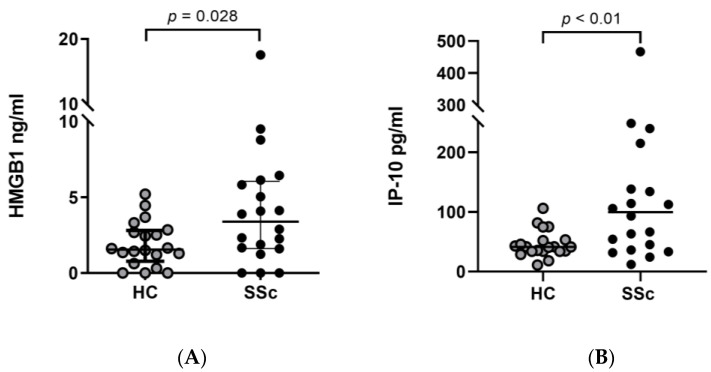
serum levels of HMGB1 and IP-10. (**A**) HMGB1 serum levels in SSc patients (*n* = 20) are significantly higher than in healthy controls (*n* = 20) *p* = 0.028. (**B**) IP-10 serum levels in SSc patients (*n* = 20) are significantly higher than in healthy controls (*n* = 20) *p* = 0.009. HMGB1 and IP-10 levels are depicted in ng/mL. HC: Healthy control; SSc: Systemic sclerosis.

## Data Availability

The data underlying this article will be shared on reasonable request to the corresponding author.

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
