# Peer review of "Release of High-Mobility Group Box-1 after a Raynaud’s Attack Leads to Fibroblast Activation and Interferon-γ Induced Protein-10 Production: Role in Systemic Sclerosis Pathogenesis"

_antioxidants, 2023, doi:10.3390/antiox12040794_

Round 1

Reviewer 1 Report

In their manuscript, Yehya Al-Adwi et al. conducted a translational study through which they investigated whether a Raynaud’s attack can promote the release of HMGB1, a nuclear factor released by apoptotic and necrotic cells after oxidative stress, and used an in vitro model in order to understand the inflammatory/fibrotic effects of HMGB1 on primary human dermal fibroblasts. The authors demonstrated that, after a Raynaud attack, only definitive SSc patients present a significant increase in HMGB1 release compared to patients with primary Raynaud and healthy controls, and that HMGB1 stimulation leads to fibroblast activation and upregulation of IFN- inducible genes such as IP-10, which was suppressed after the blockade of the RAGE-axis. The study is original, interesting and the manuscript is overall well written.

Specific comments:

- To demonstrate the in vitro effects of HMGB1 on human dermal fibroblasts, the authors measured inflammatory, profibrotic, and IFN-inducible genes by RT-qPCR, thus only assessing gene expression. In my opinion, the study design would be more complete if the authors could confirm their results also at the protein level, by performing a western blot or an immunofluorescence analysis.

- I think the style of the references throughout the text does not reflect that of the magazine. Please check.

Author Response

Dear reviewer,

Kindly refer to the attachment.

Kind regards,

Yehya

Reviewer 2 Report

This manuscript reports the release of HMGB-1 in SSc patients after a cold challenge to induce a Raynaud’s attack. The authors  observed higher levels of HMGB-1 after the cold challenge in the blood of SSc patients. They also observed higher levels of HMGB-1 and IP-10 in an independent cohort of SSc patients. Finally, they described the pro-inflammatory and pro-fibrotic effects  of recombinant HMGB-1 on dermal fibroblasts. Although the cohorts are small in number, the work is relevant, innovative and should be of interest for readers. 

Here are concerns :

- During experiments carried out in vitro on fibroblasts, it should be ensured that the recombinant protein HMGB-1 is free of endotoxins. Indeed, contamination could also explain the results obtained. 

- The authors use HMG-B1 protein of bovine origin. Why this choice?

- The cellular model used for the in vitro experiments (fibroblasts) does not seem to be the most relevant to illustrate the pathogenic role of HMG-B1 in RP. Indeed, human endothelial cells would seem more appropriate.

Author Response

(The authors gave the same response as above.)

Reviewer 3 Report

General comments:

The authors reported a possible mechanism in the pathogenesis of early phase of SSc through the Raynaud's phenomenon. It is very interest to elucidate the pathogenesis in SSc. Further minor elucidations will be needed to publish in the Journal as described below.

Specific comments:

1. I would like to know the PRP patients group in detail. It was obscure between PRP and SSc patients. What was clinical symptomatical difference between them?

2. What was the initial mechanism in response to coldness attack leading to molecular chain reaction?

3. What was medication except nefedipine, loprost, and bosentan? Only three drugs?

4. In Abstract, CTGF had no explanation.

5. In Table 2, SSc patients had 75% smoking habit. Did it affect on the study?

6. In Table 1 and 2, did SSc patients duplicate? If so, the background of SSc was different in the study and the study would lack integrity.

7. Serum amyloid A protein, one of acute phase proteins, reacts with RAGE. What was the role of such serum inflammatory factors in Raynaud's phenomenon?

Author Response

(The authors gave the same response as above.)

Round 2

Reviewer 1 Report

The authors have adequatelly addressed my comments. 

Reviewer 2 Report

The authors have responded to all comments/suggestions.